# Assessing the added value of linking electronic health records to improve the prediction of self-reported COVID-19 testing and diagnosis

Dylan Clark-Boucher[1], Jonathan Boss[1], Maxwell Salvatore[1,2], Jennifer A. Smith[2,3], Lars G. Fritsche[1,4,5☯]*, Bhramar Mukherjee[1,2,4,5☯¤]*

1 Department of Biostatistics, University of Michigan School of Public Health, Ann Arbor, Michigan, United States of America, 2 Department of Epidemiology, University of Michigan School of Public Health, Ann Arbor, Michigan, United States of America, 3 Survey Research Center, Institute for Social Research, University of Michigan, Ann Arbor, Michigan, United States of America, 4 Rogel Cancer Center, University of Michigan, Ann Arbor, Michigan, United States of America, 5 Center for Statistical Genetics, University of Michigan School of Public Health, Ann Arbor, Michigan, United States of America

☯ These authors contributed equally to this work.
¤ Current address: M4208 SPH II, 1415 Washington Heights, Ann Arbor, Michigan, United States of America
* bhramar@umich.edu (BM); larsf@umich.edu (LGF)

**Data Availability Statement:** The human research participant data underlying our analysis (survey

## Abstract

Since the beginning of the Coronavirus Disease 2019 (COVID-19) pandemic, a focus of research has been to identify risk factors associated with COVID-19-related outcomes, such as testing and diagnosis, and use them to build prediction models. Existing studies have used data from digital surveys or electronic health records (EHRs), but very few have linked the two sources to build joint predictive models. In this study, we used survey data on 7,054 patients from the Michigan Genomics Initiative biorepository to evaluate how well self-reported data could be integrated with electronic records for the purpose of modeling COVID-19-related outcomes. We observed that among survey respondents, self-reported COVID-19 diagnosis captured a larger number of cases than the corresponding EHRs, suggesting that self-reported outcomes may be better than EHRs for distinguishing COVID-19 cases from controls. In the modeling context, we compared the utility of survey- and EHR-derived predictor variables in models of survey-reported COVID-19 testing and diagnosis. We found that survey-derived predictors produced uniformly stronger models than EHR-derived predictors—likely due to their specificity, temporal proximity, and breadth—and that combining predictors from both sources offered no consistent improvement compared to using survey-based predictors alone. Our results suggest that, even though general EHRs are useful in predictive models of COVID-19 outcomes, they may not be essential in those models when rich survey data are already available. The two data sources together may offer better prediction for COVID severity, but we did not have enough severe cases in the survey respondents to assess that hypothesis in in our study.

responses and electronic health record data) cannot be publicly shared as they represent potentially identifying and sensitive patient data. However, investigators interested in accessing these restricted resources for their own research and meet the criteria for accessing sensitive data may contact the University of Michigan Precision Health's Research Scientific Facilitators at PHDataHelp@umich.edu (also see https://research. medicine.umich.edu/our-units/data-office-clinical-translational-research/data-access) to inquire about the necessary steps regarding ethics committee approval and data sharing agreement.

**Funding:** The research presented here was funded by the National Science Foundation (https://www. nsf.gov/) under grant DMS 1712933 (BM), the National Institutes of Health (https://www.nih.gov) under grant 5R01HG008773-05 (BM) and 3P30CA046592-32-S3 (BM), and the Michigan Collaborative Addiction Resources & Education System (https://micaresed.org) under grant 1UG3CA267907-01 (BM). Any opinions, findings, and conclusions or recommendations expressed in this material are those of the author(s) and do not necessarily reflect the views of the National Science Foundation. The funders had no role in study design, data collection and analysis, decision to publish, or preparation of the manuscript.

**Competing interests:** The authors have declared that no competing interests exist.

# Introduction

The Coronavirus Disease 2019 (COVID-19) pandemic, caused by the novel coronavirus SARS-CoV-2, has over the past eighteen months reached every corner of the globe. Although some countries have been successful in suppressing COVID-19 cases, the United States (US) remains a hotbed, with a staggering count of over 78 million cases and 900 thousand deaths as of February 27th, 2022 [1]. Over 2 million people have been infected in the State of Michigan alone, resulting in more than 31,000 deaths [2]. Even as the initial pandemic has waned, novel SARS-CoV-2 variants such as B.1.617.2 ("delta"), and more recently B.1.1.529 ("omicron"), have continued to spark new waves of cases worldwide [3–8].

A focal point of research has been to identify predictors of COVID-19-related outcomes, such as testing and diagnosis, and use them to build predictive models. Early in the pandemic, COVID-19 testing was targeted at those who had symptoms or were thought to be high risk [9], meaning that healthcare worker status, essential worker status, and severity of COVID-19 symptoms are all effective predictors of being tested [10,11]. Studies have also outlined predictors of COVID-19 susceptibility, including demographic features, like age and sex, along with certain comorbidities and social habits [9–12]. Non-White racial groups, especially Blacks, are disadvantaged and overrepresented among COVID-19 cases and deaths due to prevailing health disparities and being disproportionately employed as "essential workers" [9–11,13].

Predictive models of COVID-19-related outcomes have tended to rely on either electronic health records (EHRs) or survey data. In ideal cases, EHRs can paint a detailed picture of a patient's medical history, providing demographic information, anthropometrics, and longitudinal disease and procedure codes. Vaid et al. used EHRs to predict mortality and critical events on a sample of 4,098 COVID-19 patients in New York City, USA, attaining Areas Under the Receiver Operating Characteristic curve (AUCs) of 0.79 to 0.89 [14]. Feng et al. used EHRs, specifically lab results, to predict suspected COVID-19 pneumonia with an AUC of 0.94 among 32 hospital admittees in Beijing, China [15]. Using EHR-derived comorbidities and social variables, Hippisley-Cox et al. built a model explaining 74.1% of the variation in time to death among 626,656 adults hospitalized for COVID-19 post COVID-19 vaccination in the United Kingdom [16].

Though EHRs have their advantages—such as costs, detail, and sample size—they also have limitations. Indeed, what EHRs in terms of depth of medical diagnoses and procedure, they sometimes lack in depth, as non-medical lifestyle, behavioral or demographic data are typically either limited or unavailable [17,18]. Survey data, in contrast, can be more specific to the topic of interest. Recent studies have used email correspondence, mobile applications, and other digital survey tools to study COVID-19 status in relation to exposure, occupation, social habits, and many other non-medical variables [10,11,19,20]. With survey data on 3,829 adult mobile-phone application users across the United States, Allen et al. obtained an AUC of 0.79 for predicting positive COVID-19 test results using pre-test data only [10]. Surveys may also be able to identify COVID-19 cases with greater success, as EHRs tend to be system- or hospital-specific and may miss out on COVID-19 cases that were diagnosed or treated at offsite locations.

Given the differences between survey data and EHRs, it is natural to question their relative utility in model development for COVID-19-related outcomes. However, to our knowledge, a direct comparison of EHR- and survey-derived variables has not been explored on the same population for prediction of these COVID-19 related outcomes. In this study, we explicitly consider whether the addition of EHR variables to survey data can improve prediction models for COVID-19 testing and diagnosis. On a sample of 7,054 participants of Michigan Medicine biorepositories, we combine data from the Michigan Medicine COVID-19 Survey [11] with EHR-derived comorbidities and sociodemographic variables; then, using both pools of

variables, we construct prediction models for two outcomes: (1) whether survey respondents have been tested for COVID-19 and (2) whether they have been diagnosed with COVID-19 by either a physician or test, comparing model performance using AUC.

## Materials and methods

### Data

Survey data for this study came from the Michigan Medicine COVID-19 Survey, which was conducted from May 26[th] to June 23[rd] of 2020 with the goal of assessing COVID-19 risk factors among participants of Michigan Medicine biorepositories. The Michigan Medicine COVID-19 Survey contained 96 total questions and spanned several topics of interest, from basic demographic information to COVID-19 testing, diagnosis, symptoms, and exposure, as well details about health status and social habits. Respondents were asked, among other topics, about their living situation, professional work, socioeconomic status, physical health, mental health, drug use, medication use, changes in behavior due to the pandemic, perceptions of pandemic-related restrictions, and efforts to avoid contracting COVID-19. Most questions were multiple choice, though some involved a typed-in numerical response (e.g., height and weight), and many questions were either binary or used a Likert scale. A full version of the survey with branching logic was published by Wu and Hornsby et al. 2020 and is available online via the following URL: https://doi.org/10.1371/journal.pone.0246447.s004 [11]. The survey was issued by email to 50,512 participants with 8,422 complete responses (16.7% response rate). We restricted the survey data to 7,054 individuals who participated in the Michigan Genomics Initiative (MGI) study and for whom EHRs were available (Fig 1). Respondents to

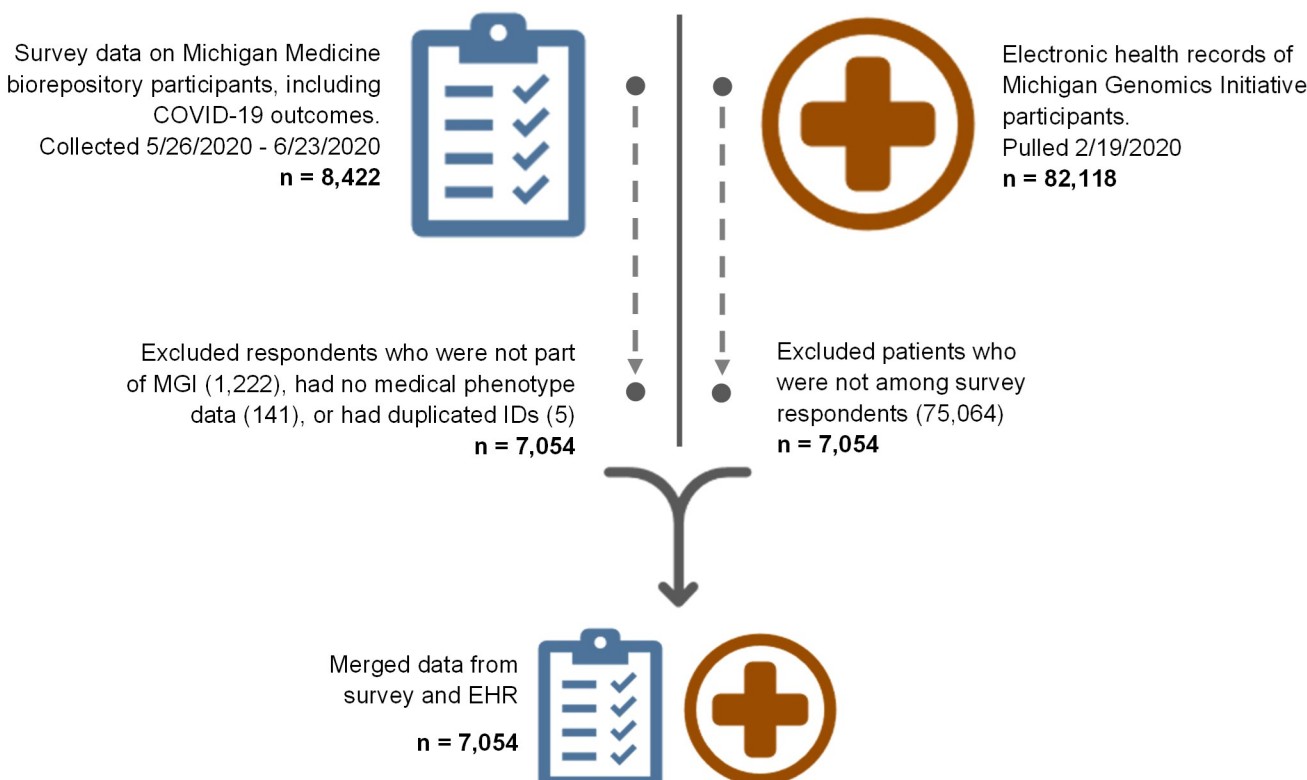

**Fig 1. Processing of COVID-19 survey data and Michigan Genomics Initiative EHRs.** For the survey-based analysis, respondents from the COVID-19 survey were limited to those with relevant EHR from MGI. The desired EHR variables were then merged with the survey data for those patients.

the survey who had participated in other repositories besides MGI were not considered, as we did not have approval to access the individual level data of these participants. EHRs used were based on a data pull from February 19th, 2020. This date was chosen to precede the initial COVID-19 outbreak in Michigan, which began in March of 2020 [2], so that the data would include medical conditions and other variables that existed strictly before the COVID-19 pandemic, as opposed to those that may have been due to a COVID-19 infection or the pandemic itself.

Data were collected according to the Declaration of Helsinki principles [21]. MGI study participants' consent forms and protocols were reviewed and approved by the University of Michigan Medical School Institutional Review Board (IRB ID HUM00180294 and HUM00155849). Opt-in written informed consent was obtained from all participants for the use of medical records and survey data. All data were fully anonymized prior to our access.

## Outcomes and variables

We constructed prediction models for two outcomes of interest: first, whether an individual was tested for COVID-19, and second, whether they were diagnosed as COVID-19 positive (by either a test or a physician) or not. Both outcomes were determined by the first two questions of the survey, which asked, (1) "Were you diagnosed with COVID-19?" and (2) "Were you tested for COVID-19 at any point in time? If so, where?" Whether each respondent was diagnosed or tested only once or multiple times was not recorded, nor was information on the specific type of test. We also considered the six variables age, sex, race/ethnicity, body mass index (BMI), education, and essential worker status as covariates, all of which have been shown to be associated with COVID-19 testing, diagnosis, or severity [9–11,13,19,22]. All prediction models contained this base set of covariates. All selections of additional predictors was carried out conditional on the covariates. In addition to these, 143 potential predictor variables were extracted for analysis from the survey data, covering a broad range of topics related to health, social habits, and experiences during the COVID-19 pandemic. Another set of 15 potential predictor variables were obtained from the EHRs, including drinking status, smoking status, and US census tract sociodemographic variables for the year 2010 based on residential address (obtained from the National Neighborhood Data Archive [23]). The 15 EHR-derived predictor variables also included seven different health condition indicator variables, which were constructed using the available phenotype codes (or "phecodes") [24,25], as well as a comorbidity score representing the sum of those from zero to seven. A mapping from phecodes to the more common International Classification of Disease codes (ICD9: 1.2, ICD10-CM: 1.2b1) is available through the PheWAS Catalog website (https://phewascatalog. org/) [24,25]. Each variable is described in detail in the supplement (S1 Table).

## Missing data

Missing data were handled with multiple imputation [26]. Under a missing at random assumption, we used predictive mean matching to create 30 multiply imputed datasets. In accordance with imputation guidelines [27,28], the number 30 was chosen to roughly coincide with proportion of incomplete cases in the full dataset—in our case, 29%.

## Statistical analysis

**Single-predictor analysis.** Both outcomes—being tested and being diagnosed—were treated as binary. For the COVID-19 testing models, we compared those who had been tested for COVID-19 at any point (1) to those who had not been (0), based on their responses first two questions of the survey. For the COVID-19 diagnosis models, we compared those who

were diagnosed with COVID-19 either by a test or by a physician (1) to those who were never diagnosed or tested (0). Respondents who reported to have self-diagnosed themselves with COVID-19 without a positive test were excluded from the diagnosis models, as these cases were unconfirmed [11].

We began by evaluating the association between each potential predictor and each outcome, while adjusting for age, sex, race/ethnicity, BMI, education, and essential worker status as covariates. ([Fig 2A]). Using Firth logistic regression, we fit the model

$$\text{logit } \Pr(Y_i = 1 \mid \vec{X}_i^T, \ \vec{C}_i^T) = \vec{X}_i^T \vec{\beta} + \vec{C}_i^T \vec{\gamma},$$

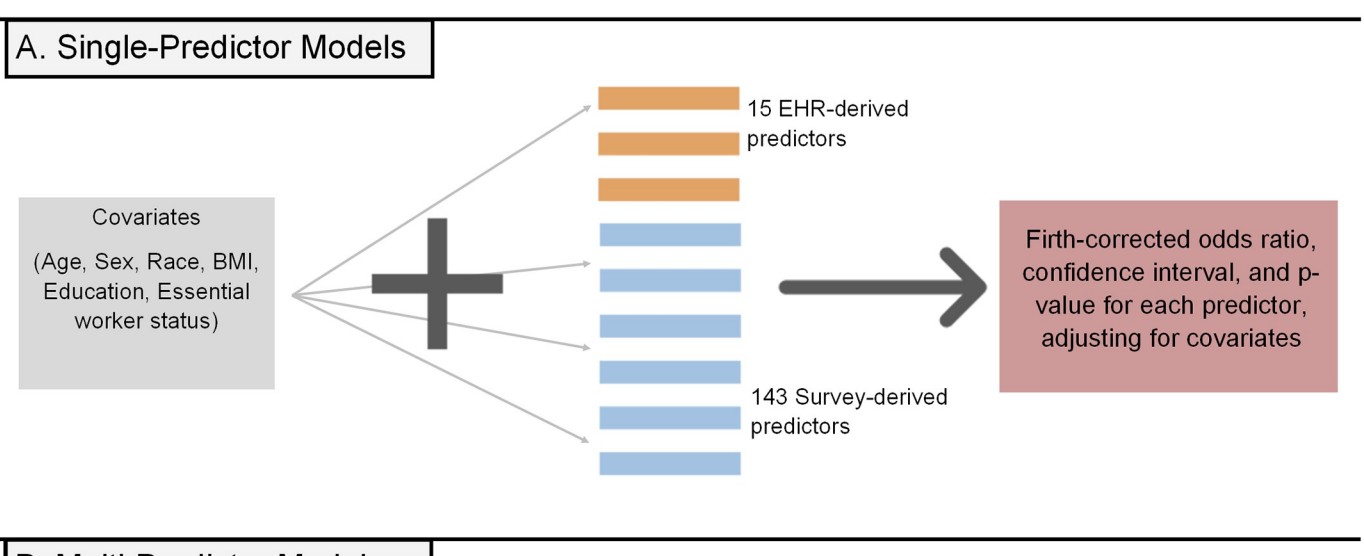

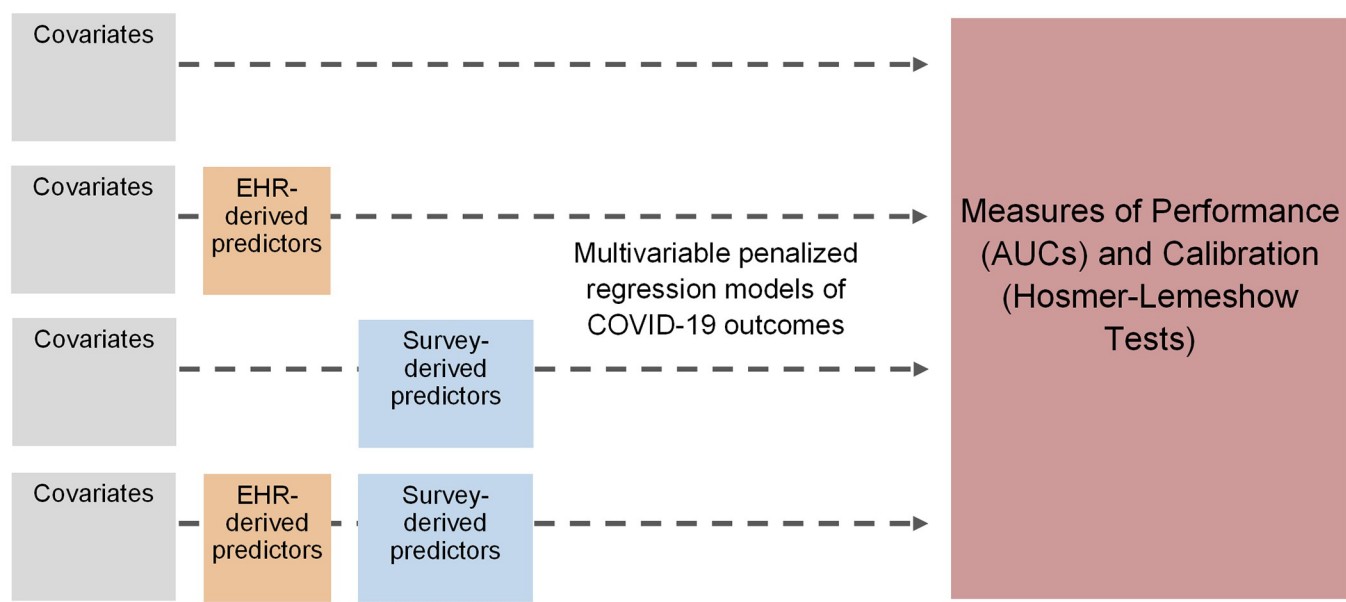

**Fig 2. Modeling of survey-reported COVID-19 outcomes.** (A) Initially, each possible predictor was tested for each outcome individually, adjusting for covariates. (B) Next, different subsets of the data were used to run penalized multivariable models, allowing for comparison between groups of variables.

where $Y_i$ is the binary outcome for the $i^{\text{th}}$ observation, $\vec{X}_i^T$ contains the value(s) of the predictor variable, and $\vec{C}_i\, T$ includes the intercept and covariates. Firth's bias-corrected odds ratios were produced for each variable and then tested for significance using the Wald test. The Firth correction is used to prevent separation issues, which can occur when large class imbalances or small sample sizes pull $\vec{\beta}$ estimates toward the extreme [29–31]. To account for multiple testing among the potential predictors, we used Bonferroni correction corresponding to 184 tests ($\alpha = 0.05/184 \approx 2.72 \times 10^{-4}$), where 184 is the total number of variables or variable factor levels tested (25 non-binary categorical variables had a total of 76 levels, excluding covariates). Last, to test the covariates on their own, we fit models for both outcomes omitting $\vec{X}_i^T$, and used the Wald test for $\vec{\gamma}$ without multiple correction ($\alpha = 0.05$). Each model was fitted on the 30 imputed datasets separately, and results were combined using Rubin's rules [32,33].

**Multi-predictor analysis.** Next, we conducted a multi-predictor analysis to compare variables from different sources for predicting each outcome (Fig 2B). We generated four "subsets" of the data for comparison: (1) the covariates only, (2) the EHR predictors and covariates, (3) the survey predictors and covariates, and (4) all variables put together. On each subset, we fit several different penalized logistic regression models for predicting the outcomes. Models included (1) a ridge penalized logistic regression, (2) a lasso penalized logistic regression, and (3) an elastic net penalized logistic regression. Ridge regression is useful in multivariable settings as a way to lessen the effects of multicollinearity and improve the precision of the model, at the cost of introducing some bias, by adding an $L_2$ penalty $\lambda_2 \sum_j \beta_j^2$ to the loss function [34]. Lasso follows in a similar vein, only with an $L_1$ penalty $\lambda_1 \sum_j |\beta_j|$ instead. This comes with the added benefit of performing variable selection in addition to shrinkage; by reducing some $\beta_j$ to exactly zero, the lasso penalty will eliminate poor predictors from the model as needed [35]. In ridge regression this is not the case, and all predictor variables are retained. Finally, elastic net regression uses the ridge and lasso penalties simultaneously to exploit the advantages of both [36]. The six covariates were included in all models and were not selected for or penalized. The tuning parameters $\lambda_1$ and $\lambda_2$ were selected by five-fold cross-validation.

We evaluated the models internally using AUC (Fig 3). Each of the 30 imputed datasets was divided 100 times into a 70–30 train-test split. At each split, the model was fitted on each of the 30 imputed training sets and evaluated on the corresponding test set, and the 30 AUCs were pooled into a single value representing performance on that split. The mean of the 100 pooled AUCs was taken as the estimated AUC of the model, and the 2.5$^{\text{th}}$ and 97.5% percentiles were used to create an empirical 95% confidence interval. We tested for a difference in AUC between two models by computing a confidence interval for their pairwise AUC differences over all 100 splits, considering the AUCs to be significantly different if the resulting interval did not contain zero. We assessed the calibration of each model by producing calibration plots corresponding to the first split, and by plotting a distribution of the Hosmer-Lemeshow test p-values of all 100 splits [37].

Finally, to check the sensitivity of all our results to social environment, we reran both our single-predictor and multi-predictor models additionally adjusting for the composite metric Neighborhood Socioeconomic Disadvantage Index (NDI), one of the US census tract variables obtained from the National Neighborhood Data Archive. NDI is computed as the average of the proportion of the census tract population below the US poverty level, the proportion unemployed, the proportion with public assistance income, and the proportion female-headed families with children. Results for the sensitivity analysis are available in the supplement (S3–S6 Tables). All analyses were completed using R version 4.1.2. Multiple imputation was performed with the package 'mice' [26]. Firth's corrected models used the 'logistf' package [38],

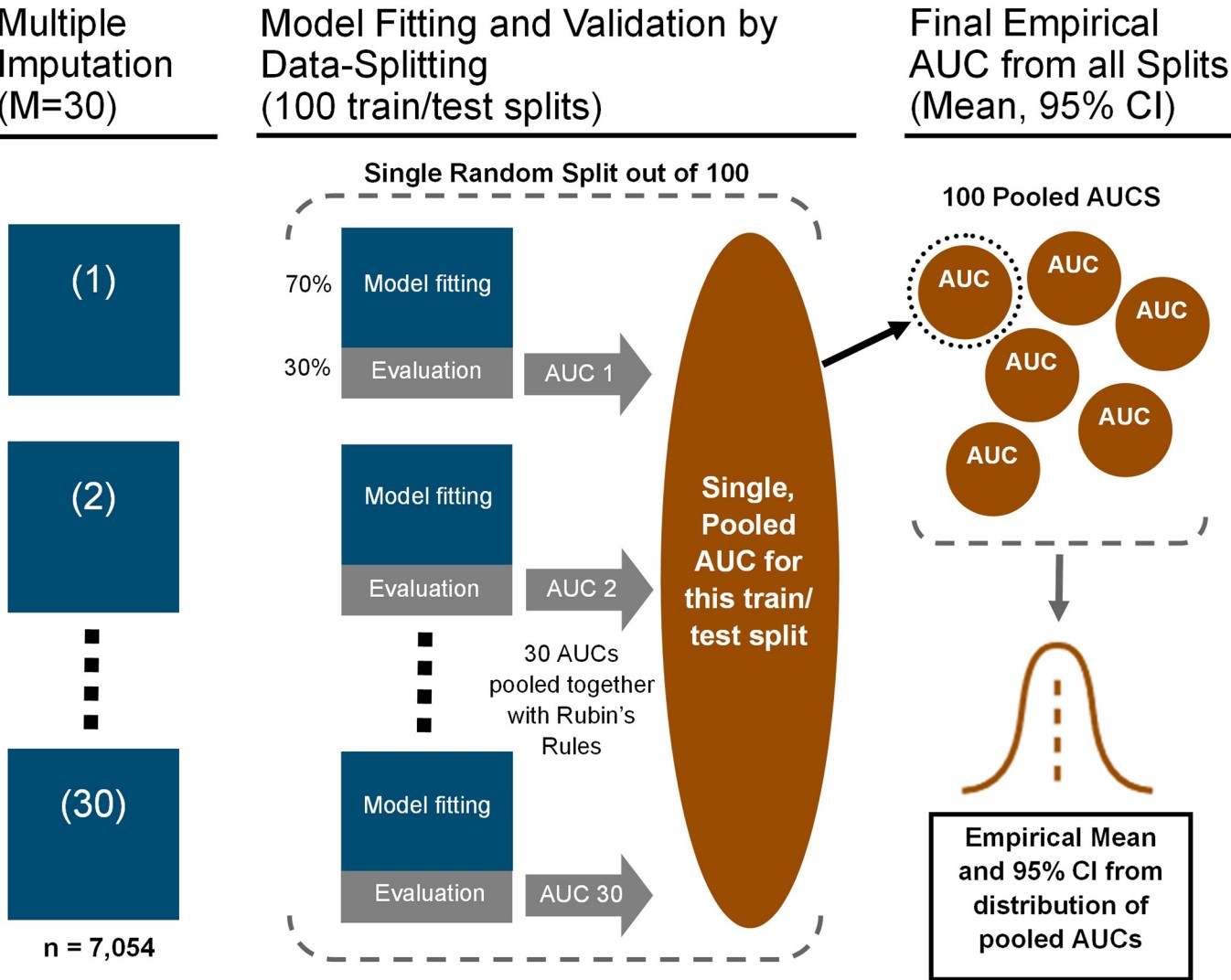

**Fig 3. Internal validation by repeated 70/30 train/test data splitting on multiply imputed datasets.** We evaluated each model internally using data splitting. For each split of the data, we pooled the results of the 30 imputed datasets into a single AUC using Rubin's Rules. The resulting 100 pooled AUCs were used to compute an empirical mean and confidence interval.

whereas the lasso, ridge, and elastic net penalized models fitted with the 'glmnet' package [39] and tuned using the 'caret' package [40]. AUCs were obtained using 'pROC' [41] and pooled across imputed datasets with 'psfmi' [42]. We produced calibration plots with 'PredictABEL' [43].

### EHR-based analysis

As a supplementary analysis, we explored how these same 15 EHR-derived variables would perform for prediction on the larger set of patients for whom only EHR data was available. Since this analysis is not directly relevant to our question and head-to-head comparison of survey versus EHR is not feasible, we relegate this analysis to the supplementary material. We used Michigan Medicine EHRs to construct cases and controls for both outcomes. To construct the COVID-19 tested cases, we retrieved data for all 15,929 patients who had obtained a

reverse transcription polymerase chain reaction (RT-PCR) test for SARS-CoV-2 at Michigan Medicine between March 10[th] and June 30[th], 2020. For COVID-19 diagnosed cases, we used the 1,193 who had tested positive, along with another 290 patients who had had COVID-19 per their EHRs but had no test results (this latter group would have included, for example, patients who were treated for COVID-19 at Michigan Medicine but were not tested there). This resulted in a total of 1,483 diagnosed cases for the analysis. These dates were chosen to align approximately with the timeline of the COVID-19 Survey, so that both analyses would capture COVID-19-related outcomes which had occurred in the same window: namely, the first four months of the pandemic in Michigan. Last, for controls, we extracted data for 30,000 random patients who were alive, were not in the tested or diagnosed groups, and had an encounter in Michigan Medicine (Inpatient, Outpatient, or Emergency) between April 23, 2012, and June 21, 2020. We used this data to fit similar models as described above and evaluated performance using AUC. A detailed description of our procedure and results is included in the supplement (S2 File).

## Results

### Descriptive statistics

Out of 7,054 survey respondents with electronic health records in MGI, only 842 (11.9%) were tested for COVID-19, 78 (1.11%) diagnosed by physician or test, and 132 (1.87%) self-diagnosed due to symptoms (Table 1). Survey respondents were 58.1 years old on average, with a standard deviation of 14.7 years, which was higher than both the tested subgroup (Mean 56.6 years, SD 14.7) and the diagnosed subgroup (Mean 49.5, SD 14.8). Their average BMI was 29.2 (SD 6.7), and they were 59.9% female. Roughly 20.1% of respondents were essential workers, compared to only 29.1% of the tested subgroup and 46.2% of the diagnosed subgroup. In terms of race-ethnicity, 6,545 (92.8%) respondents were non-Hispanic White, but only 158 (2.2%) were non-Hispanic Black. Respondents also tended to be highly educated, as 2,510

**Table 1. Descriptive statistics of covariates across survey-reported COVID-19 outcomes.**

| Variables | All (n = 7,054) | Tested (n = 842) | Diagnosed by physician or test (n = 78) | Self-diagnosed due to symptoms (n = 132) |
|---|---|---|---|---|
| **Numeric, Mean (SD)** | | | | |
| Age (Years) | 58.1 (14.7) | 56.6 (14.7) | 49.4 (14.8) | 54.9 (12.8) |
| BMI (kg/m$^2$) | 29.2 (6.72) | 29.8 (6.82) | 30.0 (6.91) | 29.1 (6.13) |
| **Categorical, No. (%)** | | | | |
| Female Sex | 4,223 (59.9%) | 542 (64.4%) | 50 (64.1%) | 89 (67.4%) |
| Essential Worker | 1,421 (20.1%) | 245 (29.1%) | 36 (46.2%) | 36 (27.3%) |
| Race / Ethnicity | | | | |
| NHB | 158 (2.24%) | 37 (4.39%) | 6 (7.69%) | 4 (3.03%) |
| NHW | 6,545 (92.8%) | 755 (89.7%) | 64 (82.1%) | 123 (93.2%) |
| Other | 261 (3.70%) | 41 (4.87%) | 6 (7.69%) | 4 (3.03%) |
| Missing | 90 (1.28%) | 9 (1.07%) | 2 (2.56%) | 1 (0.75%) |
| Education | | | | |
| ≤ High School | 1,180 (16.7%) | 173 (20.6%) | 16 (20.5%) | 23 (17.4%) |
| Associate Degree | 1,128 (16.0%) | 152 (18.1%) | 14 (17.9%) | 20 (15.2%) |
| Bachelor's Degree | 2,204 (31.2%) | 240 (28.6%) | 22 (28.2%) | 43 (32.6%) |
| Advanced Degree | 2,510 (35.6%) | 275 (32.7%) | 25 (32.1%) | 45 (34.1%) |
| Missing | 32 (0.45%) | 2 (0.24%) | 1 (1.28%) | 1 (0.76%) |

Abbreviations: BMI, Body Mass Index; NHB, non-Hispanic Black; NHW, non-Hispanic White.

(35.6%) had an advanced degree and only 1,180 (16.7%) had strictly a high school education or less.

To check for nonconformity between the survey data and EHRs, we scanned the EHRs of all respondents for a COVID-19 diagnosis prior to the date they completed the survey. Only 14 respondents had a positive diagnosis in their records, and all of them had reported so in the survey. This suggests that non-reporting of COVID-19 was not an issue and that the survey was better able to identify COVID-19 cases than the MGI EHRs, as 64 of 78 self-reported COVID-19 cases had no such diagnosis in their electronic records.

## Single-predictor models

For predicting COVID-19 testing and COVID-19 diagnosis, we used logistic regression to produce Firth bias-corrected odds ratios for every variable, adjusting for the covariates age, sex, BMI, race, education, and essential worker status (Fig 4). Of the 143 survey variables tested, only 32 were significant predictors of being tested for COVID-19 after Bonferroni correction, most of them related to overall health, such as whether the respondent has poor sleep quality (OR, 1.60 [CI, 1.34–1.92]) or has much difficulty going up stairs compared to none (OR, 2.43 [CI, 1.87–3.15]). In contrast, only three of the 15 EHR variables were significant: chronic kidney disease (OR, 1.54 [CI, 1.27–1.86]), comorbidity score (OR, 1.14 [CI, 1.08–1.20]), and respiratory conditions (OR, 1.33 [CI, 1.14–1.56]. For predicting diagnosis, seventeen survey variables were significant—some related to possible exposure, like having had a relative diagnosed with COVID-19 (OR, 8.70 [CI, 5.07–14.90]), and others related to overall health, like reporting headaches in the past six months (OR, 2.45 [CI, 1.48–4.06]), while only two of the EHR-derived variables were significant: comorbidity score (OR, 1.26 [CI, 1.08–1.47]) and respiratory conditions (OR, 2.07 [CI, 1.24–3.48]).

## Multi-predictor models

To compare the predictive power of the survey variables to the EHR variables, we ran penalized multiple logistic regression models on four different subsets of the data: the covariates alone, the covariates plus EHR variables, the covariates plus survey variables, and at last all variables put together. AUCs for all models are included in Table 2, though for succinctness we discuss the results from the elastic net models only. For predicting survey-reported COVID-19 testing, the EHR-variable model achieved a mean AUC of 0.595 across 100 different training-test splits, whereas the survey-variable and all-variable models attained mean AUCs of 0.649 and 0.648, respectively. A 95% empirical confidence interval (CI) for the AUC differences indicated that the survey-variable model was significantly more predictive than the EHR model (CI AUC$_{Survey–EHR}$, [0.031, 0.078]), the all-variable model was more predictive than the EHR model (CI AUC$_{All–EHR}$, [0.032, 0.076]), and the survey- and all-variable models were similarly predictive (CI AUC$_{All–Survey}$, [-0.001, 0.004]). In other words, showing all variables to the models at once did not improve predictions compared to using survey variables but not EHR. Results for survey-reported COVID-19 diagnosis were similar, but with generally higher AUCs: The EHR-variable model reached an average AUC of 0.709, and the survey-variable model 0.802, which was again significant at the 5% level (CI AUC$_{Survey–EHR}$, [0.015, 0.178]). The all-variable model performed nearly identically to the survey-variable model yet again (CI AUC$_{All–Survey}$, [-0.016,0.020]), with a mean AUC of 0.804. The addition of EHR-derived variables to survey-based models again seemed to offer no gain in predictive performance.

Lastly, to illustrate some of the variables these models have chosen, we report the variables selected at least 80% of the time from the elastic net models for both outcome variables. Results for the lasso regression models are included in the supplements (S8 Table). Note that, in the

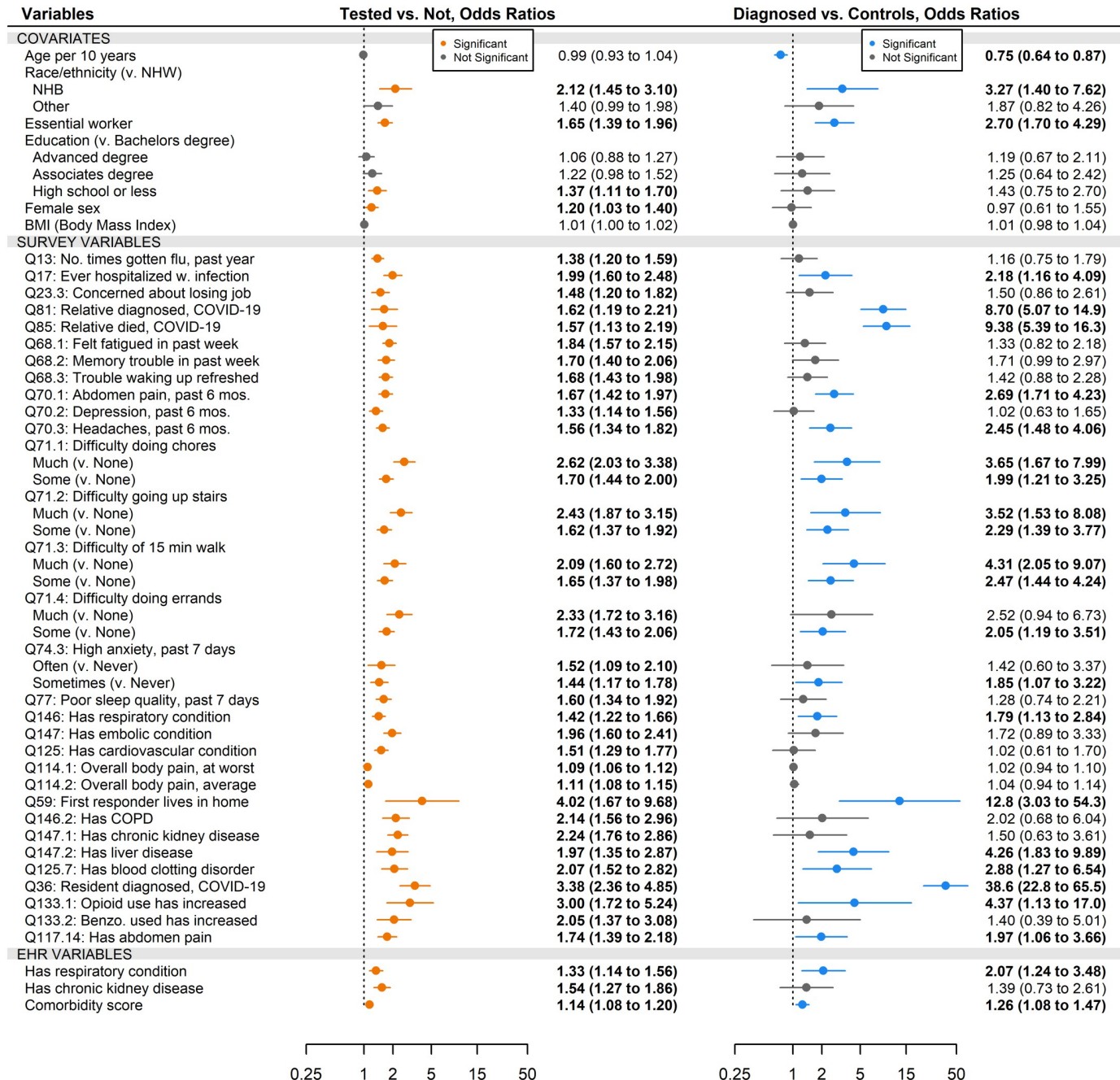

**Fig 4. Firth-corrected odds ratios for survey-reported COVID-19 outcomes, adjusted for covariates.** All odds ratios used Firth correction, were adjusted for age, sex, race/ethnicity, education, and essential worker status, and were combined across 30 multiply imputed datasets using Rubin's Rules. Significance was determined using $\alpha = 0.05$ for covariates and $\alpha = 0.05/184 \approx 2.72 \times 10^{-4}$ for predictors. For brevity, predictors are included in the figure only if they are statistically significant for at least one outcome.

present context, it would be inappropriate to provide estimated model coefficients, as they would be adjusted for different sets of selected variables and therefore have different interpretations. For predicting self-reported COVID-19 testing (Table 3), the EHR-variable models chose comorbidity score 99% of the time, a kidney disease indicator variable 98% of the time, and a respiratory disease indicator 94% of the time. In the survey-variable models, several

**Table 2. Area Under the Curve (AUC) and the corresponding 95% CI for the two COVID-19-related outcome prediction models.**

| Outcome Variable | Model Type | Mean AUC (95% Empirical CI)* | | | |
|---|---|---|---|---|---|
| | | Covariates Only | Covariates + EHR Variables | Covariates + Survey Variables | All Variables |
| Tested for COVID-19 | Lasso | 0.582 (0.552, 0.609) | 0.593 (0.569, 0.617) | 0.646 (0.621, 0.676) | 0.645 (0.619, 0.674) |
| | Ridge Regression | 0.582 (0.552, 0.609) | 0.597 (0.57, 0.623) | 0.641 (0.618, 0.67) | 0.639 (0.616, 0.668) |
| | Elastic Net | 0.582 (0.552, 0.609) | 0.595 (0.569, 0.62) | 0.649 (0.624, 0.678) | 0.648 (0.624, 0.676) |
| Diagnosed with COVID-19 | Lasso | 0.694 (0.599, 0.774) | 0.694 (0.599, 0.774) | 0.798 (0.718, 0.885) | 0.798 (0.718, 0.885) |
| | Ridge Regression | 0.694 (0.599, 0.774) | 0.713 (0.615, 0.793) | 0.812 (0.741, 0.885) | 0.821 (0.743, 0.887) |
| | Elastic net | 0.694 (0.599, 0.774) | 0.709 (0.612, 0.788) | 0.802 (0.728, 0.878) | 0.804 (0.724, 0.88) |

Mean AUC reflects the average of 100 random training test/splits, with a CI representing the 2.5th and 97.5th percentiles, respectively. The tested for COVID-19 outcome compares the tested population (1) to those not tested (0). The diagnosed with COVID-19 outcome compares those diagnosed with COVID-19 by a physician or test (1) to those not diagnosed, not tested, and not self-diagnosed (0).

Data from Michigan Medicine COVID-19 Survey and Michigan Genomics Initiative. Sample size: n = 7,054 for testing outcome models, n = 6,159 for diagnosis models.

predictors had extremely high selection rates, such as whether a person had ever been hospitalized with a viral infection (Q17, 1.00) and whether a member of their household (besides themselves) had been diagnosed with COVID-19 (Q36, 1.00). The all-variable model results look similar to the survey-variable model results, and no EHR-derived variables were selected more than 80% of the time.

For predicting self-reported COVID-19 diagnoses (Table 4), the most frequent predictor chosen in the EHR-variable models was an indicator for liver disease (0.92), the only predictor to be chosen more than 90% of the time. In the survey-variable models, the most selected variable again was whether a member of the respondent's household had been diagnosed with COVID-19 (Q36, 1.00), though similar variables—such as having had a relative diagnosed with COVID-19 (Q81, 0.82)—were popular as well. The all-variable models produced similar results to the survey-variable models, and again no EHR-derived variables were selected more than 80% of the time. Overall, selection rates tended be lower for the COVID-19 diagnosis outcome than the COVID-19 testing outcome (Table 3), a sign that the former has fewer strong predictors and requires sparser models.

## Discussion

Our aim was to evaluate whether survey-based predictive models of COVID-19-related outcomes can be improved by the addition of EHR data. Among up to 7,054 survey respondents, we analyzed two outcomes of interest—having received a COVID-19 test and having been diagnosed with COVID-19—by fitting models using EHR variables, survey variables, and then all variables combined, while using six covariates as a baseline for comparison. We observed, for both outcomes, that simultaneously including both EHR variables and survey variables led to no meaningful improvement compared to survey variables alone, with maximum AUCs around 0.65 for COVID-19 testing and 0.82 for diagnosis. In a supplementary analysis, exploring how EHR-derived variables would perform on EHR-derived outcomes, we built models on COVID-19 case control data from Michigan Medicine (S2 File). The resulting AUCs tended to be relatively high (0.75 for COVID-19 testing and 0.80 for diagnosis), suggesting that EHR-

**Table 3. Elastic net regression variable selection in models of "received COVID-19 test" outcome.**

| Proportion of Times Selected | | | | | |
|---|---|---|---|---|---|
| **EHR Variables Models** | | **Survey Variable Models** | | **All Variable Models** | |
| Comorbidity score | 0.99 | Q17. Ever Hospitalized with infection | 1.00 | Q17. Ever Hospitalized with infection | 1.00 |
| Kidney disease | 0.98 | Q36. Household member diagnosed with COVID-19 | 1.00 | Q36. Household member diagnosed with COVID-19 | 1.00 |
| Respiratory disease | 0.94 | Q147.1 Kidney disease | 1.00 | Q147.1 Kidney disease | 1.00 |
| Liver disease | 0.91 | Q68.1 Felt fatigued in past week | 1.00 | Q68.1 Felt fatigued in past week | 1.00 |
| Former smoker | 0.80 | Q70.1 Abdomen pain in past 6 months | 1.00 | Q70.1 Abdomen pain in past 6 months | 1.00 |
| | | Q70.3 Headaches in past 6 months | 1.00 | Q70.3 Headaches in past 6 months | 0.99 |
| | | Q13. No times gotten flu in past year | 0.99 | Q13. No times gotten flu in past year | 0.99 |
| | | Q125. Cardiovascular condition | 0.98 | Q125. Cardiovascular condition | 0.98 |
| | | Q146.2 COPD | 0.98 | Q146.2 COPD | 0.97 |
| | | Q147. Metabolic Condition | 0.96 | Q125.7 Blood clotting disorder | 0.95 |
| | | Q125.7 Has cardiovascular condition | 0.95 | Q147. Metabolic condition | 0.94 |
| | | Q59.1 Police officer lives in home | 0.95 | Q59.1 Police officer lives in home | 0.94 |
| | | Q23.3 Concerned about losing job | 0.95 | Q114.1 Overall body pain at worst | 0.94 |
| | | Q71.1 Some difficult doing chores | 0.94 | Q23.3 Concerned about losing job | 0.93 |
| | | Q71.1 Much difficulty doing chores | 0.94 | Q71.1 Much difficulty doing chores | 0.93 |
| | | Q114.1 Overall body pain at worst | 0.94 | Q71.1 Some difficulty doing chores | 0.93 |
| | | Q133.2 Benzodiazepine use has increased | 0.88 | Q133.2 Benzodiazepine use has increased | 0.87 |
| | | Q114.2 Overall body pain on average | 0.86 | Q68.3 Trouble waking up refreshed | 0.85 |
| | | Q46. Flu shot in past year | 0.86 | Q77. Poor sleep quality, past 7 days | 0.85 |
| | | Q77. Poor sleep quality, past 7 days | 0.86 | Q114.2 Overall body pain, on average | 0.85 |
| | | Q68.3 Trouble waking up refreshed | 0.85 | Q133.1 Opioid use has increased | 0.83 |
| | | Q133.1 Opioid use has increase | 0.85 | Q46. Flu shot in past year | 0.83 |
| | | Q36.1. Lives alone | 0.83 | Q36.1. Lives alone | 0.81 |
| | | Q68.2 Memory trouble in past week | 0.81 | | |

The value shown is the proportion of times the variable was chosen in 3,000 fitted models, as models were fit on 100 train/test splits of 30 multiply imputed datasets (100x30 = 3,000). Only variables with a selection rate over 80% are included. Variable descriptions are available in the supplement (S1 Table). The tested for COVID-19 outcome compares the tested population (1) to those not tested (0). All models included the six covariates age, sex, race/ethnicity, body mass index, education level, and essential worker status, which were not selected for or penalized. Data from Michigan Medicine COVID-19 Survey and Michigan Genomics Initiative. Sample size: 7,054.

**Table 4. Elastic net regression variable selection in models of "diagnosed with COVID-19" outcome.**

| Proportion of Times Selected | | | | | |
|---|---|---|---|---|---|
| **EHR Variables Models** | | **Survey Variable Models** | | **All Variable Models** | |
| Liver disease | 0.92 | Q36. Household member diagnosed with COVID-19 | 1.00 | Q36. Household member diagnosed with COVID-19 | 1.00 |
| Respiratory disease | 0.87 | Q85. Relative died from COVID-19 | 0.85 | Q85. Relative died from COVID-19 | 0.82 |
| | | Q70.1 Abdomen pain in past 6 months | 0.82 | Q81. Relative diagnosed with COVID-19 | 0.81 |
| | | Q81. Relative diagnosed with COVID-19 | 0.82 | Q70.1 Abdomen pain in past 6 months | 0.80 |

The value shown is the proportion of times the variable was chosen in 3,000 fitted models, as models were fit on 1000 train/test splits of 30 multiply imputed datasets (100x30 = 3,000). Only variables with a selection rate over 80% are included. Variable descriptions are available in the supplement (S1 Table). The diagnosed with COVID-19 outcome compares those diagnosed with COVID-19 by a physician or test (1) to those not diagnosed, not tested, and not self-diagnosed (0). All models included the six covariates age, sex, race/ethnicity, body mass index, education level, and essential worker status, which were not selected for or penalized. Data from Michigan Medicine COVID-19 Survey and Michigan Genomics Initiative. Sample size: 6,159.

derived variables can be moderately predictive of COVID-19 outcomes in certain contexts, especially in case-control populations that are too large to be surveyed in full.

Our results also speak to the relative difficulty of modeling different COVID-19 outcomes. AUCs for predicting COVID-19 testing tended to be lower than those for predicting diagnosis, which is not surprising considering COVID-19 testing may be driven by many factors (e.g., work, travel, symptoms, anxiety), some of which we have not accounted for. In contrast, being diagnosed with COVID-19 is heavily tied to exposure to the virus, resulting in strong associations for certain survey variables. We also ran our models using self-diagnosis with COVID-19 as an outcome, but found only weak associations and poor predictive power (S6 Table). As symptoms of COVID-19 mirror common cold and flu symptoms [44], self-diagnosis due to symptoms may be inherently imprecise.

There are several limitations to our work. First, our sample is not representative of the US population, or even the Michigan population, as it was based on survey respondents who were disproportionately White, predominantly college-educated, and who tended to have chronic medical conditions. There are many reasons why this could have been–for instance, EHR databases are known to be subject to selection bias [45]. Biorepositories such as MGI are no exception, as MGI's initial recruiting pool consisted largely of Michigan Medicine surgery patients, a nonrandom subset of the population [46]. Further, nonresponse to the survey could have compounded these biases or introduced novel ones, as descriptive statistics of survey responders compared to recipients showed small differences [11]. Respondents to the survey also tended to be older than Michigan Medicine at large, and more female (S13 Table). The odds ratios we have presented should be viewed only in this context and not extrapolated to the general United States. Advanced tools such as inverse probability weighting, which can help account for some of these biases, were beyond the scope of the current work. Our relative comparison between survey- and EHR-derived variables is still fair conditional on a fixed pool of respondents.

Second, our models lacked external validation data and could only be evaluated internally Though our study evaluated prediction performance of the models by repeated data-splitting of the sample, alternative methods to avoid over-fitting, such as optimism correction [47], are also logical choices for an honest assessment of prediction. However, data-splitting has the advantage of simplicity, especially in the case of handling multiply imputed datasets, and the procedure should be sufficient for the sake of comparing relative predictive performance across multiple sets of variables on the same sample. Moreover, as the survey- and EHR-based models were based on different samples from different populations, we should be wary of comparing their AUCs directly.

Third, the small number of COVID-19 diagnosed cases in the data (78), relative to the number of features explored (up to 164), could have led to instability in the estimated model coefficients or performance. However, these concerns are lessened, though not eliminated entirely, by the fact that we applied shrinkage to the models via penalization. Moreover, the relative comparison between EHR- and survey-derived variables for prediction can be thought of as conditioning on a given sample, so the relative comparison is still of value despite this limitation of the data. A proper quantification that is broadly generalizable would require simulation studies with various effect sizes and sample sizes that is beyond the scope of the current work. There are also possible limitations to our specific choice of models. Variable selection approaches such as LASSO and elastic net can have difficulty choosing the optimal model under certain conditions: for instance, when predictors are correlated. For this reason, the proportion of times each variable was selected may not always be reflective of the true variable importance. Moreover, parametric generalized linear models in general may struggle to capture the true nature of exposure-outcome associations, especially when those associations are

complex and involves non-linearity and interactions. Future work could expand upon our findings by applying machine learning approaches that are more flexible than logistic regression, such as, for example random forests or neural networks as well as ensemble methods like the super learner [48].

Fourth, it may be argued that our comparison of survey variables to EHR variables is intrinsically unfair, as we tested 143 variables from the survey and only 15 from the EHRs. However, the breadth and number of variables is an important difference between EHR and survey data in general. If the additional variables that surveys can incorporate are predictive, then the difference in number of variables is not inhibitive to our comparison but fundamental to it. Our comparison is also only relevant to the specific outcome variables that we explored: being diagnosed with COVID-19 and being tested for it. Other COVID-19-related outcomes, such as severity of symptoms, may have stronger associations with EHR-derived comorbidities, potentially making the addition of EHR variables to survey data more impactful, but we did not have the data to address this question.

Last, it is possible that there are reporting errors in the survey, due to factors such as recall bias [49,50], or that the EHR-derived health conditions came with inherent accuracies [17,51,52]. An example of the former, having abdomen pain in the prior six months was associated with both COVID-19 testing and diagnosis, but respondents may have had difficulty recalling their pain over such a long period.

EHR- and survey-based research have both been critical to understanding COVID-19 outcomes and their risk factors. Our results should not be interpreted to mean that EHR data have no value when survey data are available, only that surveys offer access to a broader scope of features that may, in some cases, make general EHRs less vital. As the spread of COVID-19 is heavily tied to risk factors that EHR do not capture (e.g., exposure information, health behavior, and vaccination status), it remains a priority to continue investing in the collection and analysis of survey data. Moreover, while the pandemic continues, it may be worthwhile to expand the base-questionnaires received by patients at hospitals and clinics to include factors that are relevant to COVID-19 risk. Efficient access to detailed and accurate medical data is a prerequisite for studying COVID-19, and going forward, may be critical in identifying vulnerable groups.

## Supporting information

**S1 Table. Complete variable descriptions.** Each variable is named after the survey question it is derived from but does not always represent the syntax or nature of that question exactly. Certain survey questions were redefined for simplicity or interpretation.
(PDF)

**S2 Table. Description of Michigan Medicine EHR COVID-19 cohorts.** *Controls consisted of randomly selected patients who were alive at the time of extraction, who had an encounter with Michigan Medicine between April 23, 2012, and June 21, 2020, and who were not part of the other cohorts. **Tested cohort includes all patients who were tested for SARS-CoV-2 between March 10th and June 30th of 2020. Diagnosed cohort includes those who tested positive as well as those who were diagnosed by a physician during that span.
(PDF)

**S3 Table. Single-predictor model odds ratios for COVID-19 testing.** All odds ratios are Firth bias-corrected and combined from 30 multiply imputed datasets using Rubin's Rule's. †Adjustment 1: Models adjust for Age, Race/Ethnicity, Sex, BMI, Essential Worker Status, and Education as covariates. ‡Adjustment 2: Models additionally adjust for Neighborhood Disadvantage Index. *p Value statistically significant at $1 - \alpha$ level. **For covariates, $\alpha = 0.05$.

For other variables, $\alpha = 0.05 / k$, where $k = 184$ for Adjustment 1 models and $k = 183$ for Adjustment 2 models.
(PDF)

**S4 Table. Single-predictor model odds ratios for COVID-19 diagnosis.** All odds ratios are Firth bias-corrected and combined from 30 multiply imputed datasets using Rubin's Rule's. [†]Adjustment 1: Models adjust for Age, Race/Ethnicity, Sex, BMI, Essential Worker Status, and Education as covariates. [‡]Adjustment 2: Models additionally adjust for Neighborhood Disadvantage Index. [*]p value statistically significant at $1 - \alpha$ level. [**]For covariates, $\alpha = 0.05$. For other variables, $\alpha = 0.05 / k$, where $k = 184$ for Adjustment 1 models and $k = 183$ for Adjustment 2 models.
(PDF)

**S5 Table. Single-predictor model odds ratios COVID-19 self-diagnosis.** All odds ratios are Firth bias-corrected and combined from 30 multiply imputed datasets using Rubin's Rule's. [†]Adjustment 1: Models adjust for Age, Race/Ethnicity, Sex, BMI, Essential Worker Status, and Education as covariates. [‡]Adjustment 2: Models additionally adjust for Neighborhood Disadvantage Index. [*]p Value statistically significant at $1 - \alpha$ level. [**]For covariates, $\alpha = 0.05$. For other variables, $\alpha = 0.05 / k$, where $k = 184$ for Adjustment 1 models and $k = 183$ for Adjustment 2 models.
(PDF)

**S6 Table. Sensitivity of survey-based model AUCs to social environment.** [†]Adjustment 1: Models adjust for Age, Race/Ethnicity, Sex, BMI, Essential Worker Status, and Education as covariates. [‡]Adjustment 2: Models additionally adjust for Neighborhood Disadvantage Index. [*]Detailed descriptions of each outcome are available in the methods sections as well as the supplement (S1 Table).
(PDF)

**S7 Table. Comparison of survey respondents to Michigan Genomics Initiative and Michigan Medicine.** Michigan Medicine records include patients who received treatment at any point from 01/01/2000 to 07/27/2020 and were over 18 years of age. Note that many self-reported Caucasians did not report an ethnicity, only a race, and therefore the number of unknowns in the Michigan Medicine Race/Ethnicity variable is large. Acronyms: NHAA, Non-Hispanic African American, NHW, Non-Hispanic White.
(PDF)

**S8 Table. LASSO model most-selected variables.** The value shown is the proportion of times the variable was chosen in 3,000 fitted models, as models were fit on 1000 train/test splits of 30 multiply imputed datasets (100x30 = 3,000). Only variables with a selection rate over 80% are included. Variable descriptions are available in the supplement (S1 Table). The tested for COVID-19 outcome compares the tested population (1) to those not tested (0). The diagnosed with COVID-19 outcome compares those diagnosed with COVID-19 by a physician or test (1) to those not diagnosed, not tested, and not self-diagnosed (0). The self-diagnosed with COVID-19 outcome compares those who diagnosed themselves with COVID-19 without a test to those who were not self-diagnosed or formally diagnosed (0). All models included the six covariates age, sex, race/ethnicity, body mass index, education level, and essential worker status, which were not selected for or penalized. Data from Michigan Medicine COVID-19 Survey and Michigan Genomics Initiative. Sample size: 6,159–7,054.
(PDF)

**S9 Table. ENET model most-selected variables.** The value shown is the proportion of times the variable was chosen in 3,000 fitted models, as models were fit on 1000 train/test splits of 30 multiply imputed datasets (100x30 = 3,000). Only variables with a selection rate over 80% are

included. Variable descriptions are available in the supplement (S1 Table). The tested for COVID-19 outcome compares the tested population (1) to those not tested (0). The diagnosed with COVID-19 outcome compares those diagnosed with COVID-19 by a physician or test (1) to those not diagnosed, not tested, and not self-diagnosed (0). The self-diagnosed with COVID-19 outcome compares those who diagnosed themselves with COVID-19 without a test to those who were not self-diagnosed or formally diagnosed (0). All models included the six covariates age, sex, race/ethnicity, body mass index, education level, and essential worker status, which were not selected for or penalized. Data from Michigan Medicine COVID-19 Survey and Michigan Genomics Initiative. Sample size: 6,159–7,054.
(PDF)

**S10 Table. Mean penalties selected by elastic net regression models.** Lambda and alpha were selected by five-fold cross-validation on the training set of a single 70/30 train/test split.
(PDF)

**S11 Table. Mean penalties selected by ridge regression models.** Lambda and alpha were selected by five-fold cross-validation on the training set of a single 70/30 train/test split.
(PDF)

**S12 Table. Mean penalties selected by LASSO models.** Lambda and alpha were selected by five-fold cross-validation on the training set of a single 70/30 train/test split.
(PDF)

**S13 Table. Comparison of survey respondents to Michigan Genomics Initiative and Michigan Medicine.** Michigan Medicine records include patients who received treatment at any point from 01/01/2000 to 07/27/2020 and were over 18 years of age. Note that many self-reported Caucasians did not report an ethnicity, only a race, and therefore the number of unknowns in the Michigan Medicine Race/Ethnicity variable is large. Acronyms: NHAA, Non-Hispanic African American, NHW, Non-Hispanic White.
(PDF)

**S1 File. EHR Phecode descriptions and frequencies by data source.**
(XLSX)

**S2 File. Description and results for EHR-based case-control analysis.**
(PDF)

**S3 File. Calibration plots for models of survey-based models of COVID-19 outcomes.**
(PDF)

## Acknowledgments

The authors acknowledge the participants and coordinators of the Michigan Medicine Precision Health COVID-19 Survey (generated and deployed by Bethany Klunder, Amelia Krause and Cristen Willer) and the Michigan Genomics Initiative study for their role in the research, as well as the Michigan Medical School Data Office for Clinical and Translational Research for providing data storage, management, processing, and distribution services.

## Author Contributions

**Conceptualization:** Dylan Clark-Boucher, Bhramar Mukherjee.

**Data curation:** Lars G. Fritsche.

Formal analysis: Dylan Clark-Boucher.

Funding acquisition: Bhramar Mukherjee.

Methodology: Dylan Clark-Boucher, Jonathan Boss, Lars G. Fritsche, Bhramar Mukherjee.

Software: Lars G. Fritsche.

Supervision: Lars G. Fritsche, Bhramar Mukherjee.

Visualization: Dylan Clark-Boucher.

Writing – original draft: Dylan Clark-Boucher.

Writing – review & editing: Dylan Clark-Boucher, Jonathan Boss, Maxwell Salvatore, Jennifer A. Smith, Lars G. Fritsche, Bhramar Mukherjee.

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
