## [Decision Letter · Decision Letter 0]

18 Jan 2022

PONE-D-21-35566Assessing the added value of linking Electronic Health Records to improve the prediction of self-reported COVID-19 testing and diagnosisPLOS ONE

Dear Dr. Fritsche,

Thank you for submitting your manuscript to PLOS ONE. After careful consideration, we feel that it has merit but does not fully meet PLOS ONE’s publication criteria as it currently stands. Therefore, we invite you to submit a revised version of the manuscript that addresses the points raised during the review process.

We look forward to receiving your revised manuscript.

Kind regards,

Sinan Kardeş, M.D.

Academic Editor

PLOS ONE

Journal Requirements:

2. Please provide additional details regarding participant consent. If you are reporting a retrospective study of medical records, archived samples or third party data, please ensure that you have discussed whether all data were fully anonymized before you accessed them and/or whether the IRB or ethics committee waived the requirement for informed consent. If patients provided informed written consent to have data from their medical records used in research, please include this information.

4. We note you have included a table to which you do not refer in the text of your manuscript. Please ensure that you refer to Table 3 in your text; if accepted, production will need this reference to link the reader to the Table.

Reviewers' comments:

Reviewer's Responses to Questions

**Comments to the Author**

1. Is the manuscript technically sound, and do the data support the conclusions?

Reviewer #1: Partly

Reviewer #2: Yes

Reviewer #3: Yes

Reviewer #4: Yes

Reviewer #5: Yes

2. Has the statistical analysis been performed appropriately and rigorously? 

Reviewer #1: No

Reviewer #2: Yes

Reviewer #3: Yes

Reviewer #4: Yes

Reviewer #5: I Don't Know

3. Have the authors made all data underlying the findings in their manuscript fully available?

Reviewer #1: Yes

Reviewer #2: No

Reviewer #3: Yes

Reviewer #4: No

Reviewer #5: Yes

4. Is the manuscript presented in an intelligible fashion and written in standard English?

Reviewer #1: Yes

Reviewer #2: Yes

Reviewer #3: Yes

Reviewer #4: Yes

Reviewer #5: Yes

5. Review Comments to the Author

Reviewer #1: In this study, the authors used survey data on 7,054 patients from the Michigan Genomics Initiative biorepository to evaluate how well self-reported data could be integrated with electronic for the purpose of modelling COVID-19 outcomes. This is a very interesting study; it is very well-written. However, the statistical approaches taken has several flows that would need to be corrected before any conclusions can be drawn from these analyses. Please below:

1. To make the introduction more focused, it may be helpful to clarify that the focus of the study is the prediction of being tested for COVID-19 and being diagnosed with it. At the moment, there is lack of clarity what outcomes are being investigated.

2. Some description of the survey and what questions it entailed, in the methods section would be helpful. There are a lot of references made to it (which is understandable as this is the primary source of data) but not description of it presented. This makes it difficult to evaluate source of data.

3. Could the authors explain the reasons for excluding variables with missing data rather than imputing the missingness? It seems a wasteful approach that is likely to lead to biased estimates.

4. From the description of the statistical approaches, it would appear that the authors used the same variables as covariates and predictors. If this is so, please clarify how it may be possible for the same variables to be both predictors and covariates?

5. For each penalised regressions, please describe how the shrinkage penalty was selected and present these penalties for each approach.

6. Please note that 70/30 train-test split for internal validation is not a recommended approach as it is likely to lead to biased estimates. A more accurate approach to internal validation is optimism correction. Please see references below.

7. According to recent guidelines on how to develop and evaluate predictions models, as part of models’ evaluation, it is essential to provide calibration and calibration in the large.

8. Similarly, according to the guidelines, it is essential to provide sample calculation for prediction models, which was not done here.

9. From the description of the statistical approaches, it appears that the Bonferroni adjustments were applied to all variables. Does it include those variables that were included in the penalised models? If so, this is incorrect as penalised models already correct for collaterality by utilising a penalty.

References:

• Altman, D.G., et al., Prognosis and prognostic research: validating a prognostic model. Bmj, 2009. 338: p. b605.

• Steyerberg, E., Clinical Prediction Models. A practical approach to development, validation, and updating. Second Edition ed. 2019: Springer Nature Switzerland.

• Riley, R.D., et al., Calculating the sample size required for developing a clinical prediction model. Bmj, 2020. 368: p. m441.

• Collins, G.S., et al., Transparent reporting of a multivariable prediction model for individual prognosis or diagnosis (TRIPOD): the TRIPOD statement. Bjog, 2015. 122(3): p. 434-43.

• Harrell, F.E., Jr., K.L. Lee, and D.B. Mark, Multivariable prognostic models: issues in developing models, evaluating assumptions and adequacy, and measuring and reducing errors. Stat Med, 1996. 15(4): p. 361-87.

• Hui Zou, T.H., Regularization and variable selection via the elastic net. Journal of the Royal Statistical Society, 2005(67): p. 301-320.

• Riley, R.D., et al., External validation of clinical prediction models using big datasets from e-health records or IPD meta-analysis: opportunities and challenges. Bmj, 2016. 353: p. i3140.

• Moons, K.G., et al., Using the outcome for imputation of missing predictor values was preferred. J Clin Epidemiol, 2006. 59(10): p. 1092-101.

Reviewer #2: Excellent MS. Sound study design and methodology. Clear discussion of findings and fair assessment of weaknesses, particularly with respect to limited generalizability of findings due to low survey response rate (16.7%).

Reviewer #3: Research idea:The research idea is novel and relevant to current pandemic circumstances. The authors have been able to use data linkage to explore gaps in assessing risk of SARS-Cov-2 diagnosis and where it can be improved upon.

Problem Statement: The problem statement is well defined in the introduction. The research gap has been well articulated.

Case definition: The case definition utilised standardised measures (i.e RT-PCR for case diagnosis).

Outcomes definition: The outcomes are also defined.

Feature reduction techniques & Bias control: Efforts to control for multicollinearity using Ridge regression is applauded. The authors also utilised Elastic Net to combine regularisation of Ridge & lasso regression. Other feature reduction techniques are noted. There has been significant efforts to control for bias in this study.

Discussion: In the discussion, the authors make a good case for how how covid outcomes from EHR data should be interpreted. The gaps not addressed by this research has been acknowledged.

Recommendations: Recommendations for collection of more data points in order to accurately assess Covid risk is in order.

Review summary: The authors utilised technically rigorous methods to address a time-relevant research question. Kudos.

Reviewer #4: The manuscript entitled "Assessing the added value of linking Electronic Health Records to improve the

prediction of self-reported COVID-19 testing and diagnosis" tackles a revelant topic in current epidemiological research. It is well written and the objectives are clearly sated as well as the methods that were used to achieve them.

Minor comments:

1. I suggest that the authors state the reason for choosing the Michigan Genomics Initiative over other initiatives for this particular project;

2. I suggest that the data flow for survey-based and EHR-based analysis (fig 1) includes the dates in which the survey and the data pull were performed;

3. The dates of the data survey and the EHR analyses are not overlapping. It would be important to explain why and if that can lead to any different results;

4. In the "Outcomes and variables" of the methods section, the definition of "tested for COVID" and "diagnosed with COVID" should be clearly stated and the questions used in the survey transcribed. It is not clear if the test performed was a PCR or an antigen test. In addition, it is not stated if any given person was tested more than once and how this data is handled;

5. The participation rate in the survey is extremely low, which can lead to a significant (2 level) selection bias. It would be interesting to compare these individuals with the ones who have not participated in the original Michigan Medicine Repository and not with the ones that, having participated in this initiative, have not answered this particular survey.

Reviewer #5: The authors assess whether electronic health record (EHR) data can be joined to self-reported survey data to improve the modeling of COVID-19 outcomes. This results of the paper demonstrate the importance of collecting timely survey data with a broad scope of features to generate accurate risk prediction models of COVID-19 outcomes. They also show that, in certain situations, the inclusion of additional EHR predictors in these models might not significantly improve model performance.

I think that this paper provides important insights on the creation of datasets and models used to predict COVID-19 outcomes. At the same time, I believe that some portions of the paper could be improved.

Major Comments

1. I feel that some context is missing from the introduction. While the predictors contained in various data sources are discussed at length, and this is certainly an important driver of predictive model performance, a quantitative overview of existing predictive models’ is missing. A brief brief comparison of models trained on EHR and survey data would help motivate your research question.

2. Am I understanding correctly that the single-predictor models were trained on the entire survey data, and that their test results were used for variable selection in the multi-parameter models trained on subsets of the same data? If this is the case, then observations in the test sets were used to select the predictors present in the training sets. The empirical mean AUCs are likely over-optimistic for these models. The initial filtering step should be performed, wherever the survey predictors are used, as part of the model fitting procedure to avoid this issue. See Ambroise and McLachlan [2002] for a discussion on this topic. Alternatively, you could avoid this feature selection procedure altogether when training the multivariate models by relying solely on penalized logistic regressions models. The single-predictor models could then be reported as a stand-alone analysis of marginal predictor importance (adjusting for established covariates).

3. Instead of using the results of a forward selection procedure trained on the full dataset to gain insight on the features selected by the models in the training sets, why not report the proportion of times features were selected when fitting the models to the training sets? This is straightforward to compute using the forward selection, LASSO, and/or elastic-net models.

4. Related to the above: Absolute coefficient estimates produced by stepwise selection procedures tend to be upwards biased. Does the Firth-correction counteract this? If so, it might be helpful to state it.

5. I think that more discussion is needed to motivate the relevance of the EHR-based analysis to your aim of evaluating whether survey data should be augmented with EHR data. As you mention, the underlying populations for these analyses differ, and so the various models’ predictive performances can’t be compared equitably. The message of the paper might be made clearer by moving this analysis to the supplement, and only briefly mentioning in the main paper that, although these EHR-based predictors don’t appreciably improve the performance of models trained on survey data, they still have value in EHR-based case-control studies.

Minor Comments

1. By no fault of the authors, the first paragraph of the introduction is (unfortunately) outdated.

2. Five-fold cross-validation is mentioned on line 343, yet I don’t see it mentioned elsewhere. It seems that Monte Carlo cross-validation was used instead to assess model performance. Was five-fold cross-validation used to select the hyperparameters of the penalized logistic regression models on the training sets?

References

Christophe Ambroise and Geoffrey J. McLachlan. Selection bias in gene extraction on the basis of microarray gene-expression data. Proceedings of the National Academy of Sciences, 99(10):6562–6566, 2002. ISSN 0027-8424. doi: 10.1073/pnas.102102699. URL https://www.pnas.org/content/99/10/6562.

6. PLOS authors have the option to publish the peer review history of their article (what does this mean?). If published, this will include your full peer review and any attached files.

Reviewer #1: No

Reviewer #2: No

Reviewer #3: No

Reviewer #4: No

Reviewer #5: No

---

## [Author Response · Author response to Decision Letter 0]

4 Mar 2022

We are thankful to the review panel for their constructive feedback, which we are confident has resulted in substantial improvements to the originally submitted manuscript. We have attached an itemized response to each reviewer comment in the response letter. 

Some of the major highlights of the revision are:

• We have replaced our original procedure for crudely handling missing data by filtering out subjects and variables with missing data with a more principles multiple imputation approach, which not only has the advantage of increasing the study’s effective sample size, but also reducing bias. The approach is valid under data missing at random.

• In the supplements of the manuscript, we have provided comprehensive details regarding our statistical models, such as (1) the tuning parameters which were selected and (2) sample plots that assess how well the models were calibrated. 

• To identify strong predictors of COVID-19 outcomes, we provide the variables which were selected at least 80% of the time in the models as suggested by the reviewer. 

• We have moved the strictly EHR-based portion of the analysis to the supplements, as one reviewer felt that it was tangentially related to our main research question and was not necessary to describe in detail in the main text.

---

## [Decision Letter · Decision Letter 1]

30 Mar 2022

PONE-D-21-35566R1Assessing the added value of linking Electronic Health Records to improve the prediction of self-reported COVID-19 testing and diagnosisPLOS ONE

Dear Dr. Fritsche,

Thank you for submitting your manuscript to PLOS ONE. After careful consideration, we feel that it has merit but does not fully meet PLOS ONE’s publication criteria as it currently stands. Therefore, we invite you to submit a revised version of the manuscript that addresses the points raised during the review process.

We look forward to receiving your revised manuscript.

Kind regards,

Sinan Kardeş, M.D.

Academic Editor

PLOS ONE

Journal Requirements:

Reviewers' comments:

Reviewer's Responses to Questions

**Comments to the Author**

1. If the authors have adequately addressed your comments raised in a previous round of review and you feel that this manuscript is now acceptable for publication, you may indicate that here to bypass the “Comments to the Author” section, enter your conflict of interest statement in the “Confidential to Editor” section, and submit your "Accept" recommendation.

Reviewer #5: All comments have been addressed

2. Is the manuscript technically sound, and do the data support the conclusions?

Reviewer #5: Yes

3. Has the statistical analysis been performed appropriately and rigorously? 

Reviewer #5: Yes

4. Have the authors made all data underlying the findings in their manuscript fully available?

Reviewer #5: No

5. Is the manuscript presented in an intelligible fashion and written in standard English?

Reviewer #5: Yes

6. Review Comments to the Author

Reviewer #5: I’d like to thank the authors for their careful revision of the paper and for addressing my comments and questions. The manuscript was a joy to read, and the analysis is now technically sound. I have no further comments beyond those listed below.

1. Consider adding slightly more context to results of predictive models given in introduction (e.g. population information).

2. A potential limitation of using LASSO or elastic net regression for variable selection is that they are only guaranteed to asymptotically select the truly important features under stringent conditions. In particular, this metric of variable importance could be misleading if certain features are highly correlated. Have you checked the estimated correlation matrices of the variables used in your models? Otherwise, this might be worth mentioning in the discussion.

3. A limitation of the multi-predictor benchmark is the use of penalized logistic regression models. It is unlikely that parametric models are capable of capturing the true complexity of the relationships under study. Determining whether the conclusions of this study hold true when using more flexible predictive models could make for an interesting follow-up analysis.

7. PLOS authors have the option to publish the peer review history of their article (what does this mean?). If published, this will include your full peer review and any attached files.

Reviewer #5: No

---

## [Author Response · Author response to Decision Letter 1]

5 Apr 2022

Thank you for the opportunity to revise and resubmit our original research article for your consideration, titled “Assessing the added value of linking electronic health records to improve the prediction of self-reported COVID-19 testing and diagnosis” (Ref. PONE-D-21-35566R1). We are thankful to the review panel for their constructive feedback, which we are confident has resulted in substantial improvements to the originally submitted manuscript. We have attached an itemized response to the reviewer’s comments in the response letter.

The revisions can be briefly summarized as follows:

1. In the introduction, we added more context to results of previous studies, i.e., included

sample sizes and population information for each.

2. In the discussion, we provided additional information on possible limitations of variable

selection approaches such as LASSO and elastic net.

3. Finally, we discuss potential challenges of generalized linear models with complex

associations and highlight interesting follow-up analyses, such as random forests or

neural networks as well as ensemble methods like the super learner.

---

## [Decision Letter · Decision Letter 2]

13 May 2022

Assessing the added value of linking Electronic Health Records to improve the prediction of self-reported COVID-19 testing and diagnosis

PONE-D-21-35566R2

Dear Dr. Fritsche,

We’re pleased to inform you that your manuscript has been judged scientifically suitable for publication and will be formally accepted for publication once it meets all outstanding technical requirements.

Kind regards,

Sinan Kardeş, M.D.

Academic Editor

PLOS ONE

Additional Editor Comments (optional):

Reviewers' comments:

Reviewer's Responses to Questions

**Comments to the Author**

1. If the authors have adequately addressed your comments raised in a previous round of review and you feel that this manuscript is now acceptable for publication, you may indicate that here to bypass the “Comments to the Author” section, enter your conflict of interest statement in the “Confidential to Editor” section, and submit your "Accept" recommendation.

Reviewer #5: All comments have been addressed

2. Is the manuscript technically sound, and do the data support the conclusions?

Reviewer #5: Yes

3. Has the statistical analysis been performed appropriately and rigorously? 

Reviewer #5: Yes

4. Have the authors made all data underlying the findings in their manuscript fully available?

Reviewer #5: No

5. Is the manuscript presented in an intelligible fashion and written in standard English?

Reviewer #5: Yes

6. Review Comments to the Author

Reviewer #5: Thank you for addressing these minor comments. Congratulations on your interesting and important work.

7. PLOS authors have the option to publish the peer review history of their article (what does this mean?). If published, this will include your full peer review and any attached files.

Reviewer #5: No

---

## [Editor Report · Acceptance letter]

19 May 2022

PONE-D-21-35566R2 

Assessing the added value of linking Electronic Health Records to improve the prediction of self-reported COVID-19 testing and diagnosis 

Dear Dr. Fritsche:

I'm pleased to inform you that your manuscript has been deemed suitable for publication in PLOS ONE. Congratulations! Your manuscript is now with our production department. 

Kind regards, 

on behalf of

Dr. Sinan Kardeş 

Academic Editor

PLOS ONE